# Sparse Binary Compression: Towards Distributed Deep Learning with minimal Communication

## Abstract

Currently, progressively larger deep neural networks are trained on ever growing data corpora. In result, distributed training schemes are becoming increasingly relevant. A major issue in distributed training is the limited communication bandwidth between contributing nodes or prohibitive communication cost in general. To mitigate this problem we propose Sparse Binary Compression (SBC), a compression framework that allows for a drastic reduction of communication cost for distributed training. SBC combines existing techniques of communication delay and gradient sparsification with a novel binarization method and optimal weight update encoding to push compression gains to new limits. By doing so, our method also allows us to smoothly trade-off gradient sparsity and temporal sparsity to adapt to the requirements of the learning task. Our experiments show, that SBC can reduce the upstream communication on a variety of convolutional and recurrent neural network architectures by more than *four* orders of magnitude without significantly harming the convergence speed in terms of forward-backward passes. For instance, we can train ResNet50 on ImageNet in the same number of iterations to the baseline accuracy, using $\times 3531$ less bits or train it to a $1\%$ lower accuracy using $\times 37208$ less bits. In the latter case, the total upstream communication required is cut from 125 *terabytes* to 3.35 gigabytes for every participating client. Our method also achieves state-of-the-art compression rates in a Federated Learning setting with 400 clients.

## 1 Introduction

Distributed Stochastic Gradient Descent (DSGD) is a training setting, in which a number of clients jointly trains a deep learning model using stochastic gradient descent (Dean et al., 2012; Recht et al., 2011; Moritz et al., 2015). Every client holds an individual subset of the training data, used to improve the current master model. The improvement is obtained by investing computational resources to perform iterations of stochastic gradient descent (SGD). This local training produces a weight update $\Delta \mathcal{W}$ in every participating client, which in regular or irregular intervals ("communication rounds") is exchanged to produce a new master model. This exchange of weight updates can be performed indirectly via a centralized server or directly in an all-reduce operation. In both cases, all clients share the same master model after every communication round (see figure 1). In vanilla DSGD the clients have to communicate a full gradient update during every iteration. Every such update is of the same size as the full model, which can be in the range of gigabytes for modern architectures with millions of parameters (He et al., 2016; Huang et al., 2017). Over the course of multiple hundred thousands of training iterations on big datasets the total communication for every client can easily grow to more than a *petabyte*. Consequently, if communication bandwidth is limited, or communication is costly, distributed deep learning can become unproductive or even unfeasible. DSGD is a very popular training setting with many applications. On one end of the spectrum, DSGD can be used to greatly reduce the training time of large-scale deep learning models by introducing device-level data parallelism (Chilimbi et al., 2014; Zinkevich et al., 2010; Xing et al., 2015; Li et al., 2014), making use of the fact that the computation of a mini-batch gradient is perfectly parallelizable. In this setting, the clients are usually embodied by hardwired high-performance computation units (i.e. GPUs in a cluster) and every client performs one iteration of SGD per communication round. Since

Figure 1: One communication round of DSGD: a) Clients synchronize with the server. b) Clients compute a weight update independently based on their local data. c) Clients upload their local weight updates to the server, where they are averaged to produce the new master model.

communication is high-frequent in this setting, bandwidth can be a significant bottleneck. On the other end of the spectrum DSGD can also be used to enable privacy-preserving deep learning (Shokri & Shmatikov, 2015; McMahan et al., 2016). Since the clients only ever share weight updates, DSGD makes it possible to train a model from the combined data of all clients without any individual client having to reveal their local training data to a centralized server. In this setting the clients typically are embedded or mobile devices with low network bandwidth, intermittent network connections, and an expensive mobile data plan. In both scenarios, the communication cost between the individual training nodes is a limiting factor for the performance of the whole learning system. For the synchronous distributed training scheme described above, the total amount of bits communicated by every client during training is given by

$$\mathtt{b}_{total} \in \mathcal{O}(\ \underbrace{N_{iter} \times f}_{\text{\# communication rounds}}\ \times\ \underbrace{|\Delta\mathcal{W}_{\neq 0}| \times (\bar{\mathtt{b}}_{pos} + \bar{\mathtt{b}}_{val})}_{\text{\# bits per communication}}\ \times\ \underbrace{K}_{\text{\# receiving nodes}}\ ) \qquad (1)$$

where $N_{iter}$ is the total number of training iterations (forward-backward passes) every client performs, $f$ is the communication frequency, $|\mathcal{W}_{\neq 0}|$ is the sparsity of the weight update, $\bar{\mathtt{b}}_{pos}$, $\bar{\mathtt{b}}_{val}$ are the average number of bits required to communicate the position and the value of the non-zero elements respectively and $K$ is the number of receiving nodes (if $\mathcal{W}$ is dense, the positions of all weights are predetermined and no position bits are required).

Substantial research has gone into the effort of reducing the amount of communication necessary between the clients via lossy compression schemes. Using the systematic of equation 1, we can organize prior approaches into three different groups:

**Sparsification** methods restrict weight updates to modifying only a small subset of the parameters, thus reducing $|\Delta\mathcal{W}_{\neq 0}|$. Strom (2015) presents an approach (later modified by Tsuzuku et al. (2018)) in which only gradients with a magnitude greater than a certain predefined threshold are sent to the server. All other gradients are aggregated into a residual. This method achieves compression rates of up to 3 orders of magnitude on an acoustic modeling task. In practice however, it is hard to choose appropriate values for the threshold, as it may vary a lot for different architectures and even different layers. Instead of using a fixed threshold to decide what gradient entries to send, Aji & Heafield (2017) use a fixed sparsity rate. They only communicate the fraction $p$ entries of the gradient with the biggest magnitude, while also collecting all other gradients in a residual. At a sparsity rate of $p = 0.001$ their method slightly degrades the convergence speed and final accuracy of the trained model. Lin et al. (2017) present modifications to the work of Aji et al. which close this performance gap. These modifications include using a curriculum to slowly increase the amount of sparsity in the first couple communication rounds and applying momentum factor masking to overcome the problem of gradient staleness. Their method achieves compression rates ranging from $\times 270$ to $\times 600$ on different architectures, without slowdown in convergence speed.

**Communication delay** methods try to reduce the communication frequency $f$. McMahan et al. (2016) propose Federated Averaging to reduce the cumulative communication. In Federated Averaging, instead of communicating after every iteration, every client performs multiple iterations of SGD to compute a weight update. The authors observe that this delay of communication does not significantly harm the convergence speed in terms of local iterations and report a reduction in the number of necessary communication rounds by a factor of $\times 10$ - $\times 100$ on different convolutional and recurrent neural network architectures. In a follow-up work Konečný et al. (2016) combine this communication delay with random sparsification and probabilistic quantization. They restrict the

clients to learn random sparse weight updates or force random sparsity on them afterwards ("structured" vs "sketched" updates) and combine this sparsification with probabilistic quantization. While their method also combines communication delay with (random) sparsification and quantization, and achieves good compression gains for one particular CNN and LSTM model, it also causes a major drop in convergence speed and final accuracy.

**Dense quantization** methods try to reduce the amount of value bits $\bar{\mathrm{b}}_{val}$. Different quantization methods have been proposed that reduce the bit-width of the gradients to ternary (Wen et al., 2017), binary (Seide et al., 2014; Bernstein et al., 2018) or arbitrary (Alistarh et al., 2017) bitwidths. While these are theoretically well-founded and come with strong convergence guarantees, they are also limited to a maximum compression rate of $\times 32$, compared to the regular 32-bit encoding.

## 2 ON THE ACCUMULATION OF GRADIENT INFORMATION

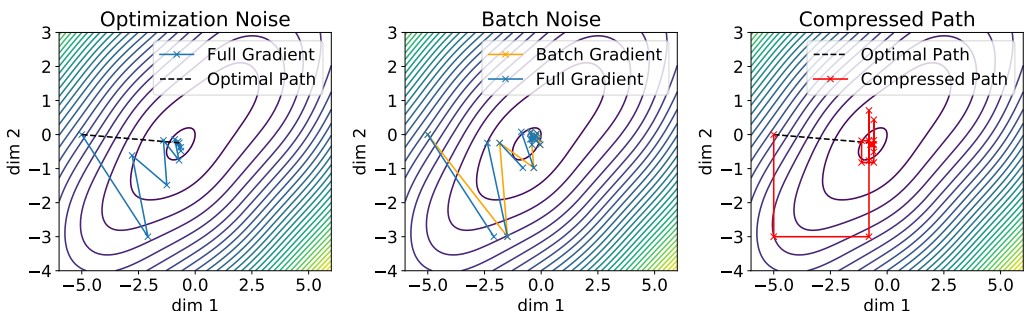

Figure 2: Sources of noise in SGD (illustration): Left: Optimization noise, caused by Gradient Descent overshooting. Bouncing between the walls of the ravine results in negatively correlated noise. Middle: Batch noise, caused by the batch loss being only a noisy approximation of the full empirical loss. Right: The compressed path converges equally fast, but requires only half of the information to be communicated.

Communication delay and sparsification methods as described above already achieve impressive compression rates, however the phenomenon underlying their successes is still only poorly understood. We present a new information-theoretic perspective that is based on the observation that both of these approaches achieve compression by accumulating gradient information locally before sending it to the server. In the case of communication delay all gradients are accumulated uniformly for a fixed amount of iterations, while in the case of sparsification methods they are accumulated non-uniformly until they exceed some fixed or adaptive threshold. In both cases the rate of compression is proportional to the number of steps that the updates are being delayed on average.

Consider now the optimization path $\Delta \mathcal{W}_1, .., \Delta \mathcal{W}_T$ taken by SGD on the loss-surface between some initialization point $\mathcal{W}_0$ and the model $\mathcal{W}_T = \mathcal{W}_0 + \sum_{t=1}^{T} \Delta \mathcal{W}_t$ trained for $T$ iterations. Following this path, we can model the changes occurring to any individual weight in the network $w$ as a noisy stochastic process via

$$\Delta w^t = s^t + n^t, \ t = 1, .., T \tag{2}$$

where $s^t$ denotes the deterministic signal (i.e. the true direction of the minimum), while $n^t$ denotes the noise, induced by mini-batch sampling in SGD ("batch noise") and the stochasticity of the learning process itself ("optimization noise", see figure 2 for an illustration). For the sake of simplicity, and motivated by the central limit theorem we can assume (a) that this noise $n^t$ is normally distributed at every time-step $n^t \sim \mathcal{N}(0, \sigma^2)$ with the variance being constant in time $\mathbb{V}(n^t) = \sigma^2$ for all $t = 1, .., T$. Since the optimization process has the tendency to damp noise as investigated for instance in LeCun et al. (2012) it is also reasonable to assume (b) that the noise is (negatively) self-correlated. The noise process is then given by $n^1 = N^1, n^t = \alpha n^{t-1} + N^t$, with $N^t$ normally distributed and all $N^t$ uncorrelated, $\alpha \in (-1, 0)$. Given these assumptions we can bound the variance of the accumulated parameter updates.

**Theorem 2.1.** *Under assumptions (a) and (b), the variance of the accumulated noise can be bounded by*

$$\mathbb{V}(\sum_{t=1}^{T} n^t) \leq \sigma^2(T(1+\alpha) + 1). \tag{3}$$

The proof can be found in the supplement. Theorem 2.1 directly leads us to a lower bound on the signal-to-noise ratio of the accumulated weight-updates:

**Corollary 2.1.1.** *Under assumptions (a) and (b), accumulation increases the signal-to-noise ratio from $\bar{s}/\sigma$ to*

$$SNR(\sum_{t=1}^{T} \Delta w^t) = \frac{\mathbb{E}[\sum_{t=1}^{T} s^t + n^t]}{\sqrt{\mathbb{V}[\sum_{t=1}^{T} s^t + n^t]}} \geq \frac{\sum_{t=1}^{T} s^t}{\sqrt{\sigma^2(T(1+\alpha)+1)}} \approx \frac{\sqrt{T}}{\sqrt{1+\alpha}} \frac{\bar{s}}{\sigma} \tag{4}$$

*with $\bar{s} = \frac{1}{T} \sum_{t=1}^{T} s^t$ being the signal-average over time.*

This means that a weight-update will be more informative the longer the accumulation period and the stronger the noise correlates temporally. Convergence speed will not be compromised for as long as the information content of the accumulated update is equal to the cumulative information content of the individual updates (c.f. fig. 2 (c)). This line of reasoning helps to shed light on both the successes of communication delay and gradient sparsification. In fact, it implies that both of these approaches are actually very similar in the way they affect the information flow from client to server on the individual weight level.

We find that this intuition is also verified empirically. Figure 3 shows validation errors for ResNet32 model trained on CIFAR for 60000 iterations at different levels of communication delay and gradient sparsity. We observe multiple things: 1.) The validation error remains more or less constant along the off-diagonals of the matrix where the total sparsity (i.e. the product of communication delay and gradient sparsity) is constant. 2.) The existing methods of Federated Averaging (McMahan et al., 2016) (purple) and Gradient Dropping/ DGC (Aji & Heafield, 2017; Lin et al., 2017)(yellow) are just lines in the two-dimensional space of possible compression methods. 3.) There exists a roughly triangular area of approximately constant error, *optimal compression methods lie along the hypotenuse of this triangle*. We find this behavior consistently across different model architectures, more examples can be found in the supplement. These results indicate, that communication delay and sparsification affect the convergence in a roughly multiplicative way and that there seems to exist a fixed information budged in DSGD, necessary to maintain unhindered convergence.

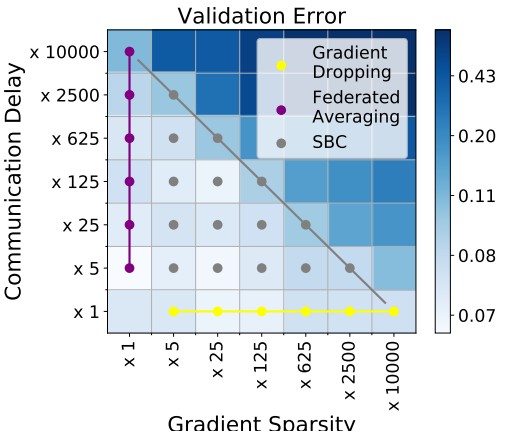

Figure 3: Validation Error for ResNet32 trained on CIFAR at different levels of temporal and gradient sparsity (the error is color-coded, brighter means lower error). The prior approaches of Gradient Dropping and Federated Averaging can be embedded in a two-dimensional compression framework.

In the following we present a framework that allows us to smoothly trade of these two types of gradient accumulation against one another. By doing so our proposed framework can adapt to the requirements of the distributed learning environment and achieve state-of-the-art compression results by reaping the benefits from both approaches.

# 3 SPARSE BINARY COMPRESSION

Inspired by our findings in the previous section, we propose Sparse Binary Compression (cf. Figure 4), to drastically reduce the number of communicated bits in distributed training. SBC makes use

of multiple compression techniques *simultaneously*[1] to reduce all multiplicative components of equation 1.

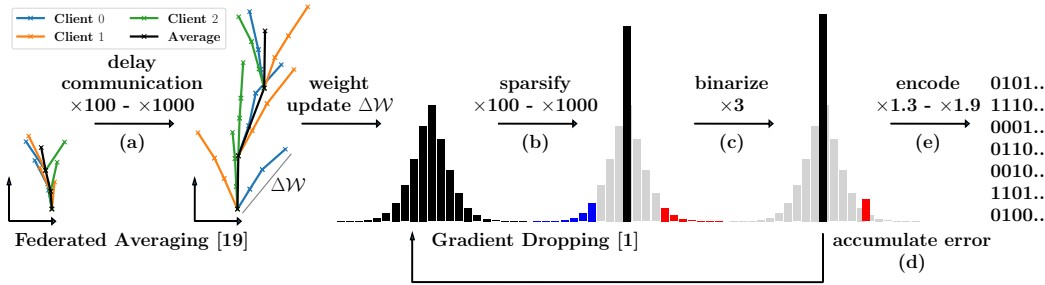

Figure 4: Step-by-step explanation of techniques used in Sparse Binary Compression: (a) Illustrated is the traversal of the parameter space with regular DSGD (left) and Federated Averaging (right). With this form of communication delay, a bigger region of the loss surface can be traversed, in the same number of communication rounds. That way compression gains of up to $\times 1000$ are possible. After a number of iterations, the clients communicate their locally computed weight updates. (b) Before communication, the weight update is first sparsified, by dropping all but the fraction $p$ weight updates with the highest magnitude. This achieves up to $\times 1000$ compression gain. (c) Then the sparse weight update is binarized for an additional compression gain of approximately $\times 3$. (d) Finally, we optimally encode the positions of the non-zero elements, using Golomb encoding. This reduces the bit size of the compressed weight update by up to another $\times 2$ compared to naive encoding.

In the following $\mathcal{W}$ will refer to the entirety of neural network parameters, while $W \in \mathcal{W}$ will refer to one specific tensor of weights. Arithmetic operations on $\mathcal{W}$ are to be understood componentwise.

**Communication Delay, Fig. 4 (a)**: We use communication delay, proposed by McMahan et al. (2016), to introduce temporal sparsity into DSGD. Instead of communicating gradients after every local iteration, we allow the clients to compute more informative updates by performing multiple iterations of SGD. These generalized weight updates are given by

$$\Delta \mathcal{W}_i = \text{SGD}_n(\mathcal{W}_i, D_i) - \mathcal{W}_i$$

where $\text{SGD}_n(\mathcal{W}_i, D_i)$ refers to the set of weights obtained by performing $n$ iterations of stochastic gradient descent on $\mathcal{W}_i$, while sampling mini-batches from the i-th client's training data $D_i$. Empirical analysis by McMahan et al. (2016) suggests that communication can be delayed drastically, with only marginal degradation of accuracy. For $n = 1$ we obtain regular DSGD.

**Sparse Binarization, Fig. 4 (b), (c)**: Following the works of Lin et al. (2017)Strom (2015)Shokri & Shmatikov (2015) and Aji & Heafield (2017) we use the magnitude of an individual weight within a weight update as a heuristic for it's importance. First, we set all but the fraction $p$ biggest and fraction $p$ smallest weight updates to zero. Next, we compute the mean of all remaining positive and all remaining negative weight updates independently. *If the positive mean $\mu^+$ is bigger than the absolute negative mean $\mu^-$, we set all negative values to zero and all positive values to the positive mean and vice versa*. The method is illustrated in figure 4 and formalized in algorithm 2. Finding the fraction $p$ smallest and biggest values in a vector $W$ requires $\mathcal{O}(|W|)$ operations, where $|W|$ refers to the number of elements in $W$ (Cormen et al., 2009). Lin et al. (2017) suggest to reduce the computational cost of this operation, by randomly subsampling from $W$. However this comes at the cost of introducing (unbiased) noise in the amount of sparsity. Luckily, in our approach communication rounds (and thus compressions) are relatively infrequent, which helps to marginalize the overhead of the sparsification. *Quantizing the non-zero elements of the sparsified weight update to the mean reduces the required value bits $\bar{b}_{val}$ from 32 to 0*. This translates to a reduction in communication cost by a factor of around $\times 3$. We can get away with averaging out the non-zero weight updates because they are relatively homogeneous in value and because we accumulate our compression errors as described in the next paragraph.

---

[1]To clarify, we have put our contributions in emphasis.

**Residual Accumulation, Fig. 4 (d)**: It is well established (Lin et al., 2017; Strom, 2015; Aji & Heafield, 2017; Seide et al., 2014) that the convergence in sparsified DSGD can be greatly accelerated by accumulating the error that arises from only sending sparse approximations of the weight updates. After every communication round, the residual is updated via

$$\mathcal{R}_\tau = \sum_{t=1}^{\tau}(\Delta\mathcal{W}_t - \Delta\mathcal{W}_t^*) = \mathcal{R}_{\tau-1} + \Delta\mathcal{W}_\tau - \Delta\mathcal{W}_\tau^*. \tag{5}$$

Error accumulation has the great benefit that no gradient information is lost (it may only become outdated or "stale"). In the context of pure sparsification residual accumulation can be interpreted to be equivalent to increasing the batch size for individual parameters (Lin et al., 2017). Moreover, we can show:

**Theorem 3.1.** *Let $\Delta W_1, .., \Delta W_T \in \mathbb{R}^n$ be (flattened) weight updates, computed by one client in the first $T$ communication rounds. Let $\Delta W_1^*, .., \Delta W_{T-1}^* \in \mathcal{S}$ be the actual weight updates, transferred in the previous rounds (restricted to some subspace $\mathcal{S}$) and $\mathcal{R}_\tau$ be the content of the residual at time $\tau$ as in equation 5. Then the orthogonal projection*

$$v = Proj_\mathcal{S}(\mathcal{R}_{T-1} + \Delta W_T) \tag{6}$$

*uniquely minimizes the accumulated error*

$$\text{err}(\Delta W_T^*) = \|\sum_{t=1}^{T}(\Delta W_t - \Delta W_t^*)\| \tag{7}$$

*in $\mathcal{S}$.* (Proof in Supplement.)

That means that the residual accumulation keeps the compressed optimization path as close as possible to optimization path taken with non-compressed weight updates.

---

**Algorithm 1:** Synchronous Distributed Stochastic Gradient Descent (DSGD)

1   **input:** initial parameters $\mathcal{W}$
2   **outout:** improved parameters $\mathcal{W}$
3   **init:** all clients $C_i$ are initialized with the same parameters $\mathcal{W}_i \leftarrow \mathcal{W}$, the initial global weight update and the residuals are set to zero $\Delta\mathcal{W}, \mathcal{R}_i \leftarrow 0$
4   **for** $t = 1, .., T$ **do**
5    **for** $i \in I_t \subseteq \{1, .., M\}$ ***in parallel*** **do**
6     Client $C_i$ does:
7     • msg $\leftarrow$ download$_{S \to C_i}$(msg)
8     • $\Delta\mathcal{W} \leftarrow$ decode(msg)

9     • $\mathcal{W}_i \leftarrow \mathcal{W}_i + \Delta\mathcal{W}$
10     • $\Delta\mathcal{W}_i \leftarrow \mathcal{R}_i + \text{SGD}_n(\mathcal{W}_i, D_i) - \mathcal{W}_i$
11     • $\Delta\mathcal{W}_i^* \leftarrow$ compress($\Delta\mathcal{W}_i$)
12     • $\mathcal{R}_i \leftarrow \Delta\mathcal{W}_i - \Delta\mathcal{W}_i^*$

13     • msg$_i \leftarrow$ encode($\Delta\mathcal{W}_i^*$)
14     • upload$_{C_i \to S}$(msg$_i$)
15    **end**
16    Server $S$ does:
17    • gather$_{C_i \to S}(\Delta\mathcal{W}_i^*)$, $i \in I_t$
18    • $\Delta\mathcal{W} \leftarrow \frac{1}{|I_t|}\sum_{i \in I_t}\Delta\mathcal{W}_i^*$
19    • $\mathcal{W} \leftarrow \mathcal{W} + \Delta\mathcal{W}$
20    • broadcast$_{S \to C_i}(\Delta\mathcal{W})$, $i = 1, .., M$
21   **end**
22   **return** $\mathcal{W}$

---

**Algorithm 2:** Sparse Binary Compression

1   **input:** tensor $\Delta W$, sparsity $p$
2   **output:** sparse tensor $\Delta W^*$
3   • val$^+ \leftarrow$ top$_{p\%}(\Delta W)$;
    val$^- \leftarrow$ top$_{p\%}(-\Delta W)$
4   • $\mu^+ \leftarrow$ mean(val$^+$); $\mu^- \leftarrow$ mean(val$^-$)
5   **if** $\mu^+ \geq \mu^-$ **then**
6    **return** $\Delta W^* \leftarrow \mu^+(W \geq \min(\text{val}^+))$
7   **else**
8    **return**
    $\Delta W^* \leftarrow -\mu^-(W \leq -\min(\text{val}^-))$
9   **end**

---

**Algorithm 3:** Golomb Position Encoding

1   **input:** sparse tensor $\Delta W^*$, sparsity $p$
2   **output:** binary message msg
3   • $\mathcal{I} \leftarrow \Delta W^*[:]_{\neq 0}$
4   • $b^* \leftarrow 1 + \lfloor\log_2(\frac{\log(\phi-1)}{\log(1-p)})\rfloor$
5   **for** $i = 1, .., |\mathcal{I}|$ **do**
6    • $d \leftarrow \mathcal{I}_i - \mathcal{I}_{i-1}$
7    • $q \leftarrow (d-1)$ div $2^{b^*}$
8    • $r \leftarrow (d-1)$ mod $2^{b^*}$
9    • msg.add($\underbrace{1, .., 1}_{q \text{ times}}, 0$, binary$_{b^*}(r)$)
10   **end**
11   **return** msg

**Optimal Position Encoding, Fig. 4 (e)**: To communicate a set of sparse binary tensors produced by SGC, we only need to transfer the positions of the non-zero elements in the flattened tensors, along with one mean value ($\mu^+$ or $\mu^-$) per tensor. Instead of communicating the absolute non-zero positions it is favorable to only communicate the distances between all non-zero elements. It is possible to show that for big values of $|W|$ and $k = p|W|$, the distances are approximately geometrically distributed with success probability equal to the sparsity rate $p$. Therefore, we can optimally encode the distances using the Golomb code Golomb (1966). Golomb encoding reduces the average number of position bits to

$$\bar{\mathrm{b}}_{pos} = \mathbf{b}^* + \frac{1}{1 - (1-p)^{2^{\mathbf{b}^*}}},\tag{8}$$

with $\mathbf{b}^* = 1 + \lfloor \log_2(\frac{\log(\phi-1)}{\log(1-p)}) \rfloor$ and $\phi = \frac{\sqrt{5}+1}{2}$ being the golden ratio. For a sparsity rate of i.e. $p = 0.01$, we get $\bar{\mathrm{b}}_{pos} = 8.38$, which translates to $\times 1.9$ compression, compared to a naive distance encoding with 16 fixed bits. *While the overhead for encoding and decoding makes it unproductive to use Golomb encoding in the situation of Strom (2015), this overhead becomes negligible in our situation due to the infrequency of weight update exchange resulting from communication delay.* The encoding scheme is given in algorithm 3, while the decoding scheme can be found in the supplement.

**Momentum Correction, Warm-up Training and Momentum Masking:** Lin et al. (2017) introduce multiple minor modifications to the vanilla Gradient Dropping method, to improve the convergence speed. We adopt momentum masking, while momentum correction is implicit to our approach. For more details on this we refer to the supplement.

Our proposed method is described in Algorithms 1, 2 and 3. Algorithm 1 describes how compression and residual accumulation can be introduced into DSGD. Algorithm 2 describes our compression method. Algorithm 3 describes the Golomb encoding. Table 1 compares theoretical asymptotic compression rates of different popular compression methods.

| | Total Bits = | Baseline | SignSGD, TernGrad , QSGD | Gradient Dropping, DGC | Federated Averaging | Sparse Binary Compression |
|---|---|---|---|---|---|---|
| | Temporal Sparsity | 100% | 100% | 100% | **0.1% - 10%** | **0.1% - 10%** |
| $\times$ | Gradient Sparsity | 100% | 100% | **0.1%** | 100% | **0.1% - 10%** |
| $\times \sum$ | Value Bits | 32 | **1 - 8** | 32 | 32 | **0** |
| | Position Bits | 0 | 0 | 16 | 0 | **8 - 14** |
| | Compression Rate | $\times \mathbf{1}$ | $\times \mathbf{4} - \times \mathbf{32}$ | $\times \mathbf{666}$ | $\times \mathbf{10} - \times \mathbf{1000}$ | $- \times \mathbf{40000}$ |

Table 1: Theoretical asymptotic compression rates for different compression methods broken down into components. Only SBC reduces all multiplicative components of the total bitsize (cf. eq. 1).

## 4 EXPERIMENTS

### 4.1 NETWORKS AND DATASETS

We evaluate our method on commonly used convolutional and recurrent neural networks with millions of parameters, which we train on well-studied data sets that contain up to multiple millions of samples. We perform experiments with client numbers ranging from 4 to 400 to cover both the distributed training and federated learning use-case.

**Image Classification:** We run experiments for LeNet5-Caffe[2] on MNIST LeCun (1998), ResNet18 and ResNet34 He et al. (2016) on CIFAR-10 and CIFAR-100 Krizhevsky et al. (2014) and ResNet50 on ILSVRC12 (ImageNet) Deng et al. (2009). For the i.i.d. setting we split the training data randomly into equally sized shards and assign one shard to every one of the clients. For the non-i.i.d. setting every client is assigned samples from only two classes of the dataset, but the amount of data still remains the same for every client. All models are trained using momentum SGD, except for LeNet5-Caffe, which is trained using the Adam optimizer Kingma & Ba (2014). Learning rate, weight intitiallization and data augmentation are as in the respective papers.

---

[2]A modified version of LeNet5 from LeCun et al. (1998) (see supplement).

| | Compression Method ⟶ | | Baseline | DGC [3] | Federated Averaging[4] | SBC (1) | SBC (2) | SBC (3) |
|---|---|---|---|---|---|---|---|---|
| 4 Clients, i.i.d. data | LeNet5-Caffe @MNIST | Accuracy | 0.9946 | 0.994 | 0.994 | 0.994 | 0.994 | 0.991 |
| | | Compression | ×1 | ×718 | ×500 | ×2071 | ×3166 | ×24935 |
| | ResNet18 @CIFAR10 | Accuracy | 0.946 | 0.9383 | 0.9279 | 0.9422 | 0.9435 | 0.9219 |
| | | Compression | ×1 | ×768 | ×1000 | ×2369 | ×3491 | × 31664 |
| | ResNet34 @CIFAR100 | Accuracy | 0.773 | 0.767 | 0.7316 | 0.767 | 0.7655 | 0.701 |
| | | Compression | ×1 | ×718 | ×1000 | ×2370 | ×3166 | ×31664 |
| | ResNet50 @ImageNet | Accuracy | 0.737 | 0.739 | 0.724 | 0.735 | 0.737 | 0.728 |
| | | Compression | ×1 | ×601 | ×1000 | ×2569 | ×3531 | ×37208 |
| | WordLSTM @PTB | Perplexity | 76.02 | 75.98 | 76.37 | 77.73 | 78.19 | 77.57 |
| | | Compression | ×1 | ×719 | ×1000 | ×2371 | ×3165 | ×31658 |
| | WordLSTM* @WIKI | Perplexity | 101.5 | 102.318 | 131.51 | 103.95 | 103.95 | 104.62 |
| | | Compression | ×1 | ×719 | ×1000 | ×2371 | ×3165 | ×31657 |

Table 2: Final accuracy/perplexity achieved on the test split and average compression rate for different compression schemes in a distributed training setting with different numbers of clients.

**Language Modeling:** We experiment with multilayer sequence-to-sequence LSTM models as described in Zaremba et al. (2014) on the Penn Treebank (PTB) Marcus et al. (1993) and Wikitext-2 corpora for next-word prediction. The PTB dataset consists of a sequence 923000 training, and 82000 validation words, while the Wikitext-2 dataset contains 2088628 train and 245569 test words. On both datasets we train a two-layer LSTM model with 650 and 200 hidden units respectively ("WordLSTM" / "WordLSTM*") with tied weights between encoder and decoder as described in Inan et al. (2016). The training data is split into consecutive subsequences of equal length, out of which we assign one to every client.

While the models we use in our experiments do not fully achieve state-of-the-art results on the respective tasks and datasets, they are still sufficient for the purpose of evaluating our compression method and demonstrate, that our method works well with common regularization techniques such as batch normalization Ioffe & Szegedy (2015) and dropout Srivastava et al. (2014). A complete description of models and hyperparameters can be found in the supplement.

## 4.2 RESULTS

We experiment with three configurations of our method: SBC (1) uses no communication delay and a gradient sparsity of 0.1%, SBC (2) uses 10 iterations of communication delay and 1% gradient sparsity and SBC (3) uses 100 iterations of communication delay and 1% gradient sparsity. Our decision for these points on the 2D grid of possible configurations is somewhat arbitrary. The experiments with SBC (1) serve the purpose of enabling us to directly compare our 0-value-bit quantization to the 32-value-bit Deep Gradient Compression (Lin et al., 2017)).

Table 2 lists compression rates and final validation accuracies achieved by different compression methods, when applied to the training of neural networks on 5 different datasets. The number of iterations (forward-backward-passes) is held constant for all methods. On all benchmarks, our methods perform comparable to the baseline, while communicating significantly less bits.

Figure 5 shows convergence speed in terms of iterations (left) and communicated bits (right) respectively for ResNet50 trained on ImageNet. The convergence speed is only marginally affected, by our different compression methods. In the first 30 epochs SBC (3) even achieves the highest accuracy, using about ×37000 less bits than the baseline. In total, SBC (3) reduces the upstream communication on this benchmark from 125 *terabytes* to 3.35 gigabytes for every participating client. After the learning rate is lowered in epochs 30 and 60 progress slows down for SBC (3) relative to the

---

[3]Lin et al. (2017) at a sparsity rate $p = 0.1\%$ without warm-up training. The gradients are encoded with Golomb encoding prior to communication.

[4]McMahan et al. (2016) at different rates of communication delay (the compression rate is equal to the communication delay).

| | Compression Method $\longrightarrow$ | | Baseline | Gradient Droping | Federated Averaging | SBC (1) | SBC (2) | SBC (3) |
|---|---|---|---|---|---|---|---|---|
| | **i.i.d. data** | | | | | | | |
| 50 | ResNet18*[5] @CIFAR10 | Accuracy | 0.9254 | 0.9167 | 0.911 | 0.921 | 0.902 | 0.906 |
| | | Compression | ×1 | ×713 | ×100 | ×2362 | ×3166 | ×31664 |
| 100 | LeNet5-Caffe @MNIST | Accuracy | 0.979 | 0.9811 | 0.967 | 0.979 | 0.9818 | 0.9536 |
| | | Compression | ×1 | ×714 | ×100 | ×2363 | ×3165 | ×31655 |
| 400 | LeNet5-Caffe @MNIST | Accuracy | 0.9758 | 0.9744 | 0.899 | 0.9731 | 0.9733 | 0.8919 |
| | | Compression | ×1 | ×714 | ×100 | ×2363 | ×3165 | ×31655 |
| | **non-i.i.d. data** | | | | | | | |
| 100 | LeNet5-Caffe @MNIST | Accuracy | 0.9506 | 0.9498 | 0.8592 | 0.9522 | 0.9583 | 0.8344 |
| | | Compression | ×1 | ×714 | ×100 | ×2363 | ×3165 | ×31655 |

Table 3: Final accuracy achieved on the test split and average compression rate for different compression schemes in a Federated learning setting with different numbers of clients.

methods which do not use communication delay. In direct comparison SBC (1) performs very similar to Gradient Dropping, while using about ×4 less bits (that is ×2569 less bits than the baseline).

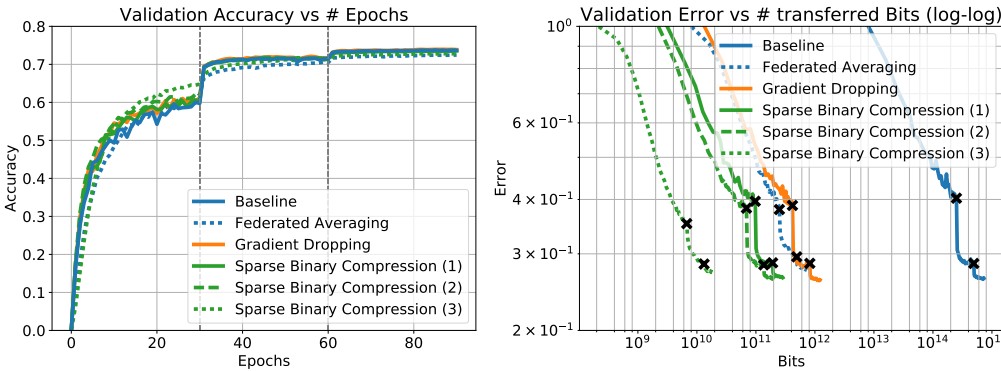

Figure 5: Left: Top-1 validation accuracy vs number of epochs. Right: Top-1 validation error vs number of transferred bits (log-log). Epochs 30 and 60 at which the learning rate is reduced are marked in the plot. ResNet50 trained on ImageNet.

Table 3 shows results for the federated learning setting with much higher numbers of clients trained on both i.i.d. and non-i.i.d. splits of data. We can see that in particular with growing numbers of clients and in the non-i.i.d. case, Federated Averaging significantly slows down the convergence and degrades the final accuracy. SBC (3) also suffers in this scenario as is also relies on 100 steps of communication delay. Conversely, our methods SBC (1) and (2) that rely more heavily on gradient sparsification perform much better in this setting and in some cases even beat the baseline. This behavior is expected, as the frequent exchange of gradient information in SBC (1) and (2) keeps all clients aligned, while they diverge further from one another for every iteration that communication is delayed in Federated Averaging.

Our experiments suggest that the distinction between the two formerly treated as separate distributed training settings of federated learning and data-parallel training is somewhat arbitrary and misleading and that better results can be achieved by combining the best approaches from both of these worlds. Contrary to the paradigm suggested in previous literature (McMahan et al., 2016), communication delay does not seem to be a well-suited approach for communication reduction in the federated learning setting. Instead our experiments demonstrate, that drastically better performance can be achieved under an even lower communication budged, if individual weight-updates are sparsified instead of delayed. On the other hand, it's easy to see that communication delay has the potential

---

[5]ResNet18* onyl has half as many convolutional filters in every layer as ResNet18.

to speed-up parallel training as it allows the individual computation devices to perform multiple steps of SGD without interruption. Our experiments with 4 clients demonstrate that introducing communication delay into data parallel training is not harmful to the convergence of the model in terms of training iterations.

## 5  CONCLUSION

The gradient information for training deep neural networks with SGD is highly redundant (see e.g. Lin et al. (2017)). We exploit this fact to the extreme by combining 3 powerful compression strategies and are able to achieve compression gains of up to *four orders* of magnitude with only a slight decrease in accuracy. More fundamentally, we present theoretical and empirical evidence suggesting that the formerly treated as separate compression methods of communication delay and gradient sparsification in fact can be viewed as two very similar forms of gradient delay that affect the convergence speed in a roughly multiplicative way. Based on this insight we propose a framework that is able to reap the benefits from both compression approaches and can smoothly adapt to communication-constraints in the learning environment, such as network bandwidth and latency and (SGD-)computation time as well as temporal inhomogeneities therein. This leads to advantages in both federated learning and data-parallel training of deep neural networks. We would like to highlight, that in no case we did modify the hyperparameters of the respective baseline models to accommodate our method. This demonstrates that our method is easily applicable. Note however that an extensive hyperparameter search could further improve the results. Furthermore, our findings in sections 2 and 4 indicate that even higher compression rates are possible if we adapt communication delay and gradient sparsity to the particular training objective. It remains an interesting direction of further research to identify heuristics and theoretical insights that can help to find the optimal balance and thus guide sparsity towards optimality.

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

## 6 SUPPLEMENT

### 6.1 MOMENTUM CORRECTION, WARM-UP TRAINING AND MOMENTUM MASKING:

Lin et al. introduce multiple minor modifications to the vanilla Gradient Dropping method. With these modifications they achieve up to around 1% higher accuracy compared to Gradient Dropping on a variety of benchmarks. Those modifications include:

*Momentum correction*: Instead of adding the raw gradient to the residuum, the momentum-corrected gradient is added. This is used implicitly in our approach, as our weight updates are already momentum-corrected.
*Warm-up Training*: The sparsity rate is increased exponentially from 25% to 0.1% in the first epochs. We find that warm-up training can indeed speed-up convergence in the beginning of training, but ultimately has no effect on the final accuracy of the model. We therefore omit warm up training in our experiments, as it adds an additional hyperparameter to the method, without any real benefit.
*Momentum Masking*: To avoid stale momentum from carrying the optimization into a wrong direction after a weight update is performed, Lin et al. suggest to set the momentum to zero for updated weights. We adopt momentum correction in our method.

### 6.2 GOLOMB POSITION DECODING

Algorithm 4 describes the decoding of a binary sequence produced by Golomb Position Encoding (see main paper). Since the shapes of all weight-tensors are known to both the server and all clients, we can omit the shape information in both encoding and decoding.

---

**Algorithm 4:** Golomb Position Decoding

1 **input:** binary message msg, bitsize $\mathbf{b}^*$, mean value $\mu$
2 **output:** sparse tensor $\Delta W^*$
3 **init:** $\Delta W^* \leftarrow 0 \in \mathbb{R}^n$
4 $\bullet$ $i \leftarrow 0; q \leftarrow 0; j \leftarrow 0$
5 **while** $i < size(\text{msg})$ **do**
6      **if** $\text{msg}[i] = 0$ **then**
7          $\bullet$ $j \leftarrow j + q2^{\mathbf{b}^*} + \text{int}_{\mathbf{b}^*}(\text{msg}[i+1], .., \text{msg}[i+\mathbf{b}^*]) + 1$
8          $\bullet$ $\Delta W_j^* \leftarrow \mu$
9          $\bullet$ $q \leftarrow 0; i \leftarrow i + \mathbf{b}^* + 1$
10      **else**
11          $\bullet$ $q \leftarrow q + 1; i \leftarrow i + 1$
12      **end**
13 **end**
14 **return** $\Delta W^*$

---

## 6.3 MODEL SPECIFICATION

Below, we describe the neural network models used in our experiments. Table 4 list the training hyperparameters that were used.

| | Experiment | Iterations | Batchsize | LR | LR Decay | Optimizer |
|---|---|---|---|---|---|---|
| 4 Clients | LeNet5-Caffe @MNIST | 2000 | 128×4 | 0.001 | - | Adam |
| | ResNet18 @CIFAR10 | 36000 | 32 × 4 | 0.1 | 0.1 @ ep 40 and 80 | Momentum SGD |
| | ResNet34 @CIFAR100 | 36000 | 32 × 4 | 0.1 | 0.1 @ ep 40 and 80 | Momentum SGD |
| | ResNet50 @ImageNet | 900000 | 32 × 4 | 0.1 | 0.1 @ ep 30 and 60 | Momentum SGD |
| | WordLSTM @PTB | 53000 | 5 × 4 | 20 | decay 0.25 if loss has not decreased | SGD |
| | WordLSTM* @WIKI | 120000 | 5 × 4 | 20 | decay 0.25 if loss has not decreased | SGD |
| 50 | Resnet18* @CIFAR10 | 23000 | 4×50 | 0.1 | 0.1 @ ep 40 and 80 | Momentum SGD |
| 100 | LeNet5-Caffe @MNIST | 2500 | 8×100 | 0.001 | - | Adam |
| 400 | LeNet5-Caffe @MNIST | 2500 | 2×400 | 0.001 | - | Adam |

Table 4: Hyperparameters used for our experiments in sections 2 and 4.

**LeNet5-Caffe**: The model specification can be downloaded from the Caffe MNIST tutorial page: `https://github.com/BVLC/caffe/blob/master/examples/mnist/lenet_train_test.prototxt`. (Features convolutional layers, fully connected layers, pooling.)

**ResNet18, ResNet32, ResNet50**: We use the implementation from the official PyTorch repository: `https://github.com/pytorch/examples/tree/master/imagenet`. (Features skip-connections, batch-normalization.)

**WordLSTM**: We use the implementation from the official PyTorch repository (configuration "medium"): `https://github.com/pytorch/examples/tree/master/word_language_model`. (Features trainable word-embeddings, multilayer LSTM-cells, dropout.)

## 6.4 PROOF OF THEOREM 2.1.

*Proof.* Since

$$n^t = \alpha n^{t-1} + N^t = \alpha(\alpha n^{t-2} + N^{t-1}) + N^t = \alpha^2 n^{t-2} + \alpha N^{t-1} + N^t$$
$$= \alpha^\tau n^{t-\tau} + \sum_{i=0}^{\tau-1} \alpha^i N^{t-i} \qquad (9)$$

it holds that

$$\text{cov}(n^{t-\tau}, n^t) = \text{cov}(n^{t-\tau}, \alpha^\tau n^{t-\tau} + \sum_{i=0}^{\tau-1} \alpha^i N^{t-i}) = \alpha^\tau \sigma^2 + \sum_{i=0}^{\tau-1} \alpha^i \underbrace{\text{cov}(n^{t-\tau}, N^{t-i})}_{=0} = \alpha^\tau \sigma^2$$
$$(10)$$

With equation equation 10 it follows that

$$\mathbb{V}(\sum_{t=1}^{T} n^t) = \sum_{t_1=1}^{T}\sum_{t_2=1}^{T} \text{cov}(n^{t_1}, n^{t_2}) \tag{11}$$

$$= \underbrace{\sum_{t=1}^{T} \text{cov}(n^t, n^t)}_{T\sigma^2} + 2\underbrace{\sum_{t=1}^{T-1} \text{cov}(n^t, n^{t+1})}_{\alpha(T-1)\sigma^2} + 2\underbrace{\sum_{t=1}^{T-2} \text{cov}(n^t, n^{t+2})}_{\alpha^2(T-2)\sigma^2} + .. + 2\underbrace{\text{cov}(n^1, n^T)}_{\alpha^{T-1}(1)\sigma^2}$$

$$\tag{12}$$

For negatively correlated noise $\alpha \in (-1, 0)$ we can bound this term by

$$\mathbb{V}(\sum_{t=1}^{T} n^t) = \sigma^2(T + 2\sum_{\tau=1}^{T-1} \alpha^\tau(T - \tau)) \tag{13}$$

$$= \sigma^2(T + 2\frac{\alpha^{T+1} - \alpha^2 T + \alpha T - \alpha}{(\alpha - 1)^2}) \tag{14}$$

$$= \sigma^2(T + 2\underbrace{\frac{(\alpha - \alpha^2)}{(\alpha - 1)^2}}_{\leq \frac{1}{2}\alpha} T + 2\underbrace{\frac{\alpha^{T+1} - \alpha}{(\alpha - 1)^2}}_{\leq \frac{1}{2}}) \tag{15}$$

$$\leq \sigma^2(T(1 + \alpha) + 1) \tag{16}$$

$\square$

## 6.5 PROOF OF THEOREM 3.1.

*Proof.* It holds that

$$\text{err}(\mathcal{R}_{T-1} + \Delta W_T) = \|\sum_{t=1}^{T} \Delta W_t - \sum_{t=1}^{T-1} \Delta W_t^* - \mathcal{R}_{T-1} - \Delta W_T\| = 0. \tag{17}$$

Since $\mathcal{S}$ is a metric subspace, the projection

$$\Delta W_T^* = \text{Proj}_{\mathcal{S}}(\mathcal{R}_{T-1} + \Delta W_T) \tag{18}$$

uniquely solves the minimization problem in $\mathcal{S}$. $\square$

## 6.6 ADDITIONAL RESULTS

Figure 6 shows validation error for WordLSTM trained on PTB at different levels of gradient sparsity and temporal sparsity. The total sparsity, defined as the product of temporal and gradient sparsity remains constant along the diagonals of the matrix. We observe that different forms of sparsity perform best during different stages of training. Phrased differently, this means that there is not one optimal sparsity setup, but rather sparsity needs to be adapted to the current training phase to achieve optimal compression.

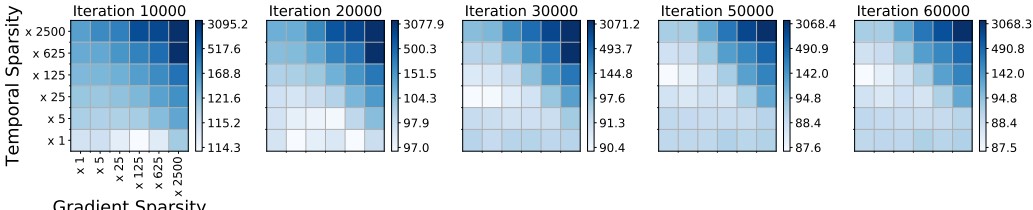

Figure 6: Perplexity for different levels of gradient sparsity and temporal sparsity at different stages of training. WordLSTM trained on PTB.

