# OpenReview forum: "Sparse Binary Compression: Towards Distributed Deep Learning with minimal Communication"
_ICLR.cc/2019/Conference_

### Official Review · AnonReviewer3 · 2018-10-19
**Good results, however, I have questions about the algorithm**

**Rating:** 5
**Confidence:** 4

**Review:**

In the paper, the authors combine the federated method, sparse compression, quantization and propose Sparse Binary Compression method for deep learning optimization.  Beyond previous methods, the method in this paper achieves excellent results in the experiments. The paper is written very clearly and easy to follow.

The following are my concerns,
1. In the introduction, the authors emphasize that there is a huge compression in the upstream communication. How about the downstream communication, I think the server should also send gradients to clients. The averaged gradient is not compressed anymore.

2. I think the method used in the paper is not federated learning. Federated learning averages the models from multiple clients. however, in the paper, the proposed methods are averaging gradients instead. It is called local updates, and is a well-known tradeoff between communication and computation in the convex optimization.

3. I want to point out that the similar local update (federated learning) technique has already explored, and proved not work well. In [1] the authors showed that deploying the local update simply may lead to divergence. Therefore, the iterations of the local update are constrained to be very small. e.g. less than 64.  Otherwise, it leads to divergence. I also got similar results in my experience.  The temporal sparsity in the paper looks very small. I am curious about why it works in this paper.

4. Another issue is the results in the experiments. It is easy to find out that resnet50 can get 76.2% on Imagenet according to [2]. However, the baseline is 73.7% in the paper.  I didn't check the result for resnet18 on cifar10 or resnet34 on cifar 100, because people usually don't use bottleneck block for cifars.

5. In Table 2, Federated average always has worse results than other compared methods. Could you explain the reason?  If using federated average is harmful to the accuracy, it should also affect the result of the proposed method.


[1] Zhang, Sixin, Anna E. Choromanska, and Yann LeCun. "Deep learning with elastic averaging SGD." Advances in Neural Information Processing Systems. 2015
[2]https://github.com/D-X-Y/ResNeXt-DenseNet

---

### Official Review · AnonReviewer2 · 2018-11-02
**I do not see this paper having enough contribution and novelty to be accepted**

**Rating:** 3
**Confidence:** 4

**Review:**

The paper once again looks at the problem of reducing the communication requirement for implementing the distributed optimization techniques, in particular, SGD. This problem has been looked at from multiple angles by many authors. And although there are many unanswered questions in this area, I do not see the authors providing any compelling contribution to answering those questions. A big chunk of the paper is devoted to expressing some shallow theorems, which in some cases I do not even see their importance or connection to the main point of the paper; see my comments below. In terms of the techniques for reducing the communication burden, the authors seem to just put all the other approaches together with minimal novelty.

More detailed comments:
- I do not really understand what the authors mean by noise damping. I would appreciate it if they could clarify that point. This seems to be a very important point as the model they propose for the noise in the process is basically based on this notion. It is a great failure on the authors' part that such a crucial notion in their paper is not clearly described.
- The model that is proposed for noise is too strong and too simplistic. Do you guys have any evidence to back this up?
- Theorem 2.1 is not a theorem. The result is super shallow and relatively trivial.
- In corollary 2.1 it seems that no matter what the randomness in the system is, the algorithm is going to converge to the same solution. This is not true even for the non-strongly convex objectives, let alone the non-convex problems where there are so many stationary solutions and whatnot.
- With regards to Fig 3 (and other related figures in the appendix) and the discussion on the multiplicative nature of compression: The figure does not seem to suggest multiplicative nature in all the regimes. It seems to hold in high compression/ low-frequency communication regime. But on the other side of the spectrum, it does not seem to hold very strongly.
- The residual accumulation only applies when all the nodes update in all the iterations. I do not believe this would generalize to the federated learning, where nodes do not participate in all the updates. I do not know if the authors have noted this point in their federated learning experiments.
- Theorem 3.1 is very poorly stated. Other that than it is shallow and in my opinion irrelevant. What is the argument in favor of the authors' thought that could be built based on the result of Theorem 3.1?
- One major point that is missing in the experiments (and probably in the experiments in other papers on the same topic) is to see how much do all these compressions affect the speed of learning in different scenarios in realistic scenarios? Note that in realistic scenarios many things other than communication could affect the convergence time.

---

### Official Review · AnonReviewer1 · 2018-11-03
**important problem, limited novelty, significance not clearly established because reporting focuses on means (bit/communication metric) not ends (optimization time)**

**Rating:** 6
**Confidence:** 4

**Review:**

Lowering costs for communicating weights between workers is an important intermediate goal for distributed optimization, since presumably it can limit the parallelization achievable once available bandwidth is saturated.  This work reports reasonable approaches to try to overcome this through a mix of techniques, though none in particular seem especially novel or surprising.  For example, the abstract claims a novel binarization method, but what is described in the paper does not seem especially novel (e.g. zero the negative weights and replace positives with their mean, if negative's mean < positive's mean, else vice versa); but more importantly, the experiments don't explore/support why this approach is any better (or when worse) than other schemes.

To its credit, the paper provides experiments data (ImageNet and Cifar, not just MNIST) and models (e.g. ResNet50) that can support reasonable claims of being representative of modern optimization tasks.  What the paper is most lacking, though, is a clear and convincing argument that the large bit compression rates claimed actually lead to significant time speedups of the resulting optimization.  The paper seems to just assume that lowering communication costs is inherently good and this goodness is proportional to the rate of compression.  But as Table 3 shows, there IS some degrading in accuracy for this reduction in communication overhead.  Whether this is worth it depends critically on whether the lower overhead actually allows optimization to speedup significantly, but the time of training seems to not be mentioned anywhere in this paper.  Thus, in its current form, this paper does not clearly establish the significance and limits of their approach.  Given that the novelty does not appear high, the value of this current paper is mainly as an engineering analysis of some design tradeoffs.  And viewed that way, this paper is a bit disappointing in that tradeoffs are not acknowledged/examined much. E.g. readers cannot tell from these experiments when the proposed approach will fail to work well -- the limitations are not clearly established (all results provided are cast in positive light).

---

### Public Comment · (anonymous) · 2018-10-26
**Just stack previous techniques**

Hi authors
     when I read this paper, you just stack several techniques together and present the report. Looks like an experiment report,
and I don't think this paper is of high novelty.
     Or can you point out the technical contribution?
Thanks

---

> ### Author Response · Authors · 2018-10-29
> **Answer to Comment**
>
> We thank you for your comment and that you show interest in our paper. We believe that we made an important contribution by demonstrating that the distinction between the two formerly treated as separate worlds of Federated Learning and Parallel Training is somewhat arbitrary and misleading and that better results can be achieved by combining the best approaches from both of these worlds. For instance, contrary to the paradigms suggested in previous literature, communication delay is not a well-suited approach for communication reduction in the Federated Learning setting. Much higher compression gains are achievable if sparsification is applied instead. On the other hand, communication delay can speed-up parallel training as it allows the individual computation devices to perform multiple steps of SGD without interruption. Our proposed method can adapt smoothly to these different settings and achieves much higher compression gains than previously reported (e.g. ×37208 on ImageNet).
> Moreover, it can also adapt to and perform optimally under different constrains that limit the communication structure, such as network bandwidth and latency, (SGD-)computation time, as well as temporal inhomogeneities therein.
> On top of that we propose a novel binarization method and Golomb encoding. Combined they can make up for another ×6 compression on top of what is achieved by communication delay and sparsification alone without harming the convergence speed.

---

### Meta-Review · Area_Chair1 · 2018-12-14
**Lack of novelty and strong empirical results; no rebuttal**

**Confidence:** 5
**Recommendation:** Reject

**Metareview:**

This paper proposes a sparse binary compression method for distributed training of neural networks with minimal communication cost. Unfortunately,  the proposed approach is not novel, nor supported by strong experiments. The authors did not provide a rebuttal for reviewers' concerns.